# Interpreting and improving natural-language processing (in machines) with natural language-processing (in the brain)

**Mariya Toneva**
Neuroscience Institute
Department of Machine Learning
Carnegie Mellon University
mariya@cmu.edu

**Leila Wehbe**
Neuroscience Institute
Department of Machine Learning
Carnegie Mellon University
lwehbe@cmu.edu

## Abstract

Neural networks models for NLP are typically implemented without the explicit encoding of language rules and yet they are able to break one performance record after another. This has generated a lot of research interest in interpreting the representations learned by these networks. We propose here a novel interpretation approach that relies on the only processing system we have that does understand language: the human brain. We use brain imaging recordings of subjects reading complex natural text to interpret word and sequence embeddings from 4 recent NLP models - ELMo, USE, BERT and Transformer-XL. We study how their representations differ across layer depth, context length, and attention type. Our results reveal differences in the context-related representations across these models. Further, in the transformer models, we find an interaction between layer depth and context length, and between layer depth and attention type. We finally hypothesize that altering BERT to better align with brain recordings would enable it to also better understand language. Probing the altered BERT using syntactic NLP tasks reveals that the model with increased brain-alignment outperforms the original model. Cognitive neuroscientists have already begun using NLP networks to study the brain, and this work closes the loop to allow the interaction between NLP and cognitive neuroscience to be a true cross-pollination.

## 1   Introduction

The large success of deep neural networks in NLP is perplexing when considering that unlike most other NLP approaches, neural networks are typically not informed by explicit language rules. Yet, neural networks are constantly breaking records in various NLP tasks from machine translation to sentiment analysis. Even more interestingly, it has been shown that word embeddings and language models trained on a large generic corpus and then optimized for downstream NLP tasks produce even better results than training the entire model only to solve this one task (Peters *et al.*, 2018; Howard and Ruder, 2018; Devlin *et al.*, 2018). These models seem to capture something generic about language. What representations do these models capture of their language input?

Different approaches have been proposed to probe the representations in the network layers through NLP tasks designed to detect specific linguistic information (Conneau *et al.*, 2018; Zhu *et al.*, 2018;

---

Code available at https://github.com/mtoneva/brain_language_nlp

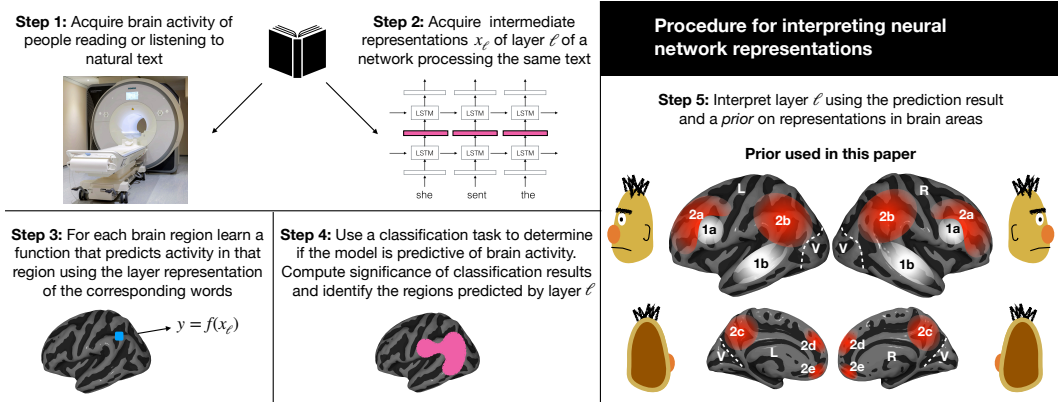

Figure 1: Diagram of approach and prior on brain function. The prior was constructed using the results of Lerner *et al.* (2011): regions in group 1 (white) process information related to isolated words and word sequences while group 2 (red) process only information related to word sequences (see Section 1.1). V indicates visual cortex. The drawing indicates the views of the brain with respect to the head. See supplementary materials for names of brain areas and full description of the methods.

Linzen *et al.*, 2016). Other approaches have attempted to offer a more theoretical assessment of how recurrent networks propagate information, or what word embeddings can represent (Peng *et al.*, 2018; Chen *et al.*, 2017; Weiss *et al.*, 2018). Most of this work has been centered around understanding the properties of sequential models such as LSTMs and RNNs, with considerably less work focused on non-sequential models such as transformers.

Using specific NLP tasks, word annotations or behavioral measures to detect if a type of information is present in a network-derived representation (such as a word embedding of an LSTM or a state vector of a transformer) can be informative. However, complex and arguably more interesting aspects of language, such as high level meaning, are difficult to capture in an NLP task or in behavioral measures. We therefore propose a novel approach for interpreting neural networks that relies on the only processing system we have that does understand language: the human brain. Indeed, the brain does represent complex linguistic information while processing language, and we can use brain activity recordings as a proxy for these representations. We can then relate the brain representations with neural network representations by learning a mapping from the latter to the former. We refer to this analysis as aligning the neural network representations with brain activity.

## 1.1   Proposed approach

We propose to look at brain activity of subjects reading naturalistic text as a source of additional information for interpreting neural networks. We use fMRI (functional Magnetic Resonance Imaging) and Magnetoencephalography (MEG) recordings of the brain activity of these subjects as they are presented text one word at a time. We present the same text to the NLP model we would like to investigate and extract representations from the intermediate layers of the network, given this text. We then learn an alignment between these extracted representations and the brain recordings corresponding to the same words to offer an evaluation of the information contained in the network representations. Evaluating neural network representations with brain activity is a departure from existing studies that go the other way, using such an alignment to instead evaluate brain representations (Wehbe *et al.*, 2014a; Frank *et al.*, 2015; Hale *et al.*, 2018; Jain and Huth, 2018).

To align a layer $\ell$ representation with brain activity, we first learn a model that predicts the fMRI or MEG activity in every region of the brain (fig. 1). We determine the regions where this model is predictive of brain activity using a classification task followed by a significance test. If a layer representation can accurately predict the activity in a brain region $r$, then we conclude that the layer shares information with brain region $r$. We can thus make conclusions about the representation in layer $\ell$ based on our prior knowledge of region $r$.

Brain recordings have inherent, meaningful structure that is absent in network-derived representations. In the brain, different processes are assigned to specific locations as has been revealed by a large array

of fMRI experiments. These processes have specific latencies and follow a certain order, which has been revealed by electrophysiology methods such as MEG. In contrast to the brain, a network-derived representation might encode information that is related to multiple of these processes without a specific organization. When we align that specific network representation with fMRI and MEG data, the result will be a decomposition of the representation into parts that correspond to different processes and should therefore be more interpretable. We can think of alignment with brain activity as a "demultiplexer" in which a single input (the network-derived representation) is decomposed into multiple outputs (relationship with different brain processes).

There doesn't yet exist a unique theory of how the brain processes language that researchers agree upon (Hickok and Poeppel, 2007; Friederici, 2011; Hagoort, 2003). Because we don't know which of the existing theories are correct, we abandon the theory-based approach and adopt a fully data-driven approach. We focus on results from experiments that use naturalistic stimuli to derive our priors on the function of specific brain areas during language processing. These experiments have found that a set of regions in the temporo-parietal and frontal cortices are activated in language processing (Lerner *et al.*, 2011; Wehbe *et al.*, 2014b; Huth *et al.*, 2016; Blank and Fedorenko, 2017) and are collectively referred to as the language network (Fedorenko and Thompson-Schill, 2014). Using the results of Lerner *et al.* (2011) we subdivide this network into two groups of areas: group 1 is consistently activated across subjects when they listen to disconnected words or to complex fragments like sentences or paragraphs and group 2 is consistently activated only when they listen to complex fragments. We will use group 1 as our prior on brain areas that process information at the level of both short-range context (isolated words) and long-range context (multi-word composition), and group 2 as a prior on areas that process long-range context only. Fig. 1 shows a simple approximation of these areas on the Montreal Neurological Institute (MNI) template. Inspection of the results of Jain and Huth (2018) shows they corroborate the division of language areas into group 1 and group 2. Because our prior relies on experimental results and not theories of brain function, it is data-driven.

We use this setup to investigate a series of questions about the information represented in different layers of neural network models. We explore four recent models: ELMo, a language model by Peters *et al.* (2018), BERT, a transformer by Devlin *et al.* (2018), USE (Universal Sentence Encoder), a sentence encoder by Cer *et al.* (2018), and T-XL (Transformer-XL), a transformer that includes a recurrence mechanism by Dai *et al.* (2019). We investigate multiple questions about these networks. Is word-level specific information represented only at input layers? Does this differ across recurrent models, transformers and other sentence embedding methods? How many layers do we need to represent a specific length of context? Is attention affecting long range or short range context?

**Intricacies**    As a disclaimer, we warn the reader that one should be careful while dealing with brain activity. Say a researcher runs a task $T$ in fMRI (e.g. counting objects on the screen) and finds it activates region $R$, which is shown in another experiment to also be active during process $P$ (e.g. internal speech). It is seductive to then infer that process $P$ is involved during task $T$. This "reverse inference" can lead to erroneous conclusions, as region $R$ can be involved in more than one task (Poldrack, 2006). To avoid this trap, we only interpret alignment between network-derived representations and brain regions if (1) the function of the region is well studied and we have some confidence on its function during a task similar to ours (e.g. the primary visual cortex processing letters on the screen or group 2 processing long range context) or (2) we show a brain region has overlap in the variance explained by the network-derived layer and by a specific process, in the same experiment. We further take sound measures for reporting results: we cross-validate our models and report results on unseen test sets.Another possible fallacy is to directly compare the performance of layers from different networks and conclude that one network performs better than the other: information is likely organized differently across networks and such comparisons are misleading. Instead we only perform controlled experiments where we look at one network and vary one parameter at a time, such as context length, layer depth or attention type.

## 1.2   Contributions

1. We present a new method to interpret network representations and a proof of concept for it.

2. We use our method to analyze and provide hypotheses about ELMo, BERT, USE and T-XL.

3. We find the middle layers of transformers are better at predicting brain activity than other layers. We find that T-XL's performance doesn't degrade as context is increased, unlike the

other models'. We find that using uniform attention in early layers of BERT (removing the pretrained attention on the previous layer) leads to better prediction of brain activity.

4. We show that when BERT is altered to better align with brain recordings (by removing the pretrained attention in the shallow layers), it is also able to perform better at NLP tasks that probe its syntactic understanding (Marvin and Linzen, 2018). This result shows a transfer of knowledge from the brain to NLP tasks and validates our approach.

## 2 Related work on brains and language

Most work investigating language in the brain has been done in a controlled experiment setup where two conditions are contrasted (Friederici, 2011). These conditions typically vary in complexity (simple vs. complex sentences), vary in the presence or absence of a linguistic property (sentences vs. lists of words) or vary in the presence or absence of incongruities (e.g. semantic surprisal) (Friederici, 2011). A few researchers instead use naturalistic stimulus such as stories (Brennan *et al.*, 2010; Lerner *et al.*, 2011; Speer *et al.*, 2009; Wehbe *et al.*, 2014b; Huth *et al.*, 2016; Blank and Fedorenko, 2017). Some use predictive models of brain activity as a function of multi-dimensional features spaces describing the different properties of the stimulus (Wehbe *et al.*, 2014b; Huth *et al.*, 2016).

A few previous works have used neural network representations as a source of feature spaces to model brain activity. Wehbe *et al.* (2014b) aligned the MEG brain activity we use here with a Recurrent Neural Network (RNN), trained on an online archive of Harry Potter Fan Fiction. The authors aligned brain activity with the context vector and the word embedding, allowing them to trace sentence comprehension at a word-by-word level. Jain and Huth (2018) aligned layers from a Long Short-Term Memory (LSTM) model to fMRI recordings of subjects listening to stories to differentiate between the amount of context maintained by each brain region. Other approaches rely on computing surprisal or cognitive load metrics using neural networks to identify processing effort in the brain, instead of aligning entire representations (Frank *et al.*, 2015; Hale *et al.*, 2018).

There is little prior work that evaluates or improves NLP models through brain recordings. Søgaard (2016) proposes to evaluate whether a word embedding contains cognition-relevant semantics by measuring how well they predict eye tracking data and fMRI recordings. Fyshe *et al.* (2014) build a non-negative sparse embedding for individual words by constraining the embedding to also predict brain activity well and show that the new embeddings better align with behavioral measures of semantics.

## 3 Approach

**Network-derived Representations**    The approach we propose in this paper is general and can be applied to a wide variety of current NLP models. We present four case-studies of recent models that have very good performance on downstream tasks: ELMO, BERT, USE and T-XL.

- ELMo is a bidirectional language model that incorporates multiple layers of LSTMs. It can be used to derive contextualized embeddings by concatenating the LSTM output layers at that word with its non-contextualized embedding. We use a pretrained version of ELMo with 2 LSTM layers provided by Gardner *et al.* (2017).

- BERT is a bidirectional model of stacked transformers that is trained to predict whether a given sentence follows the current sentence, in addition to predicting a number of input words that have been masked (Devlin *et al.*, 2018). Upon release, this recent model achieved state of the art across a large array of NLP tasks, ranging from question answering to named entity recognition. We use a pretrained model provided by Hugging Face [1].We investigate the base BERT model, which has 12 layers, 12 attention heads, and 768 hidden units.

- USE is a method of encoding sentences into an embedding (Cer *et al.*, 2018) using a task similar to Skip-thought (Kiros *et al.*, 2015). USE is able to produce embeddings in the same space for single words and passages of text of different lengths. We use a version of USE from tensorflow hub trained with a deep averaging network [2] that has 512 dimensions.

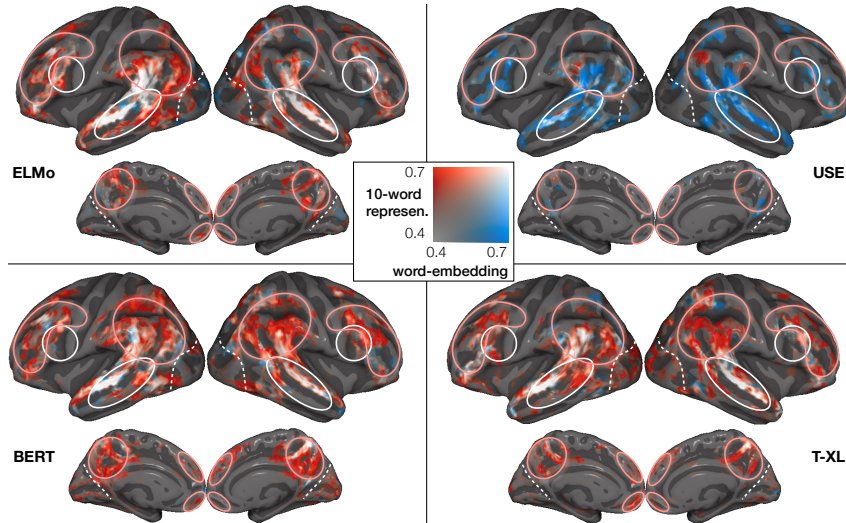

Figure 2: Comparison between the prediction performance of two network representations from each model: a 10-word representation corresponding to the 10 most recent words shown to the participant (Red) and a word-embedding corresponding to the last word (Blue). Areas in white are well predicted from both representations. These results align to a fair extent with our prior: group 2 areas (red outlines) are mostly predicted by the longer context representations while areas 1b (lower white outlines) are predicted by both word-embeddings and longer context representations.

- T-XL incorporates segment level recurrence into a transformer with the goal of capturing longer context than either recurrent networks or usual transformers (Dai *et al.*, 2019). We use a pretrained model provided by Hugging Face[1], with 19 layers and 1024 hidden units.

We investigate how the representations of all four networks change as we provide varying lengths of context. We compute the representations $x_{\ell,k}$ in each available intermediate layer ($\ell \in \{1, 2\}$ for ELMo; $\ell \in \{1, ..12\}$ for BERT; $\ell$ is the output embedding for USE; $\ell \in \{1, ..19\}$ for T-XL). We compute $x_{l,k}$ for word $w_n$ by passing the most recent $k$ words ($w_{n-k+1}, .., w_n$) through the network.

**fMRI and MEG data**  In this paper we use fMRI and MEG data which have complementary strengths. fMRI is sensitive to the change in oxygen level in the blood that is a consequence to neural activity, it has high spatial resolution (2-3mm) and low temporal resolution (multiple seconds). MEG measures the change in the magnetic field outside the skull due to neural activity, it has low spatial resolution (multiple cm) and high temporal resolution (up to 1KHz). We use fMRI data published by Wehbe *et al.* (2014b). 8 subjects read chapter 9 of *Harry Potter and the Sorcerer's stone* Rowling (2012) which was presented one word at a time for a fixed duration of 0.5 seconds each, and 45 minutes of data were recorded. The fMRI sampling rate (TR) was 2 seconds. The same chapter was shown by Wehbe *et al.* (2014a) to 3 subjects in MEG with the same rate of 0.5 seconds per word. Details about the data and preprocessing can be found in the supplementary materials.

**Encoding models**  For each type of network-derived representation $x_{\ell,k}$, we estimate an encoding model that takes $x_{\ell,k}$ as input and predicts the brain recording associated with reading the same $k$ words that were used to derive $x_{\ell,k}$. We estimate a function $f$, such that $f(x_{l,k}) = y$, where $y$ is the brain activity recorded with either MEG or fMRI. We follow previous work (Sudre *et al.*, 2012; Wehbe *et al.*, 2014b,a; Nishimoto *et al.*, 2011; Huth *et al.*, 2016) and model $f$ as a linear function, regularized by the ridge penalty. The model is trained via four-fold cross-validation and the regularization parameter is chosen via nested cross-validation.

**Evaluation of predictions**  We evaluate the predictions from each encoding model by using them in a classification task on held-out data, in the four-fold cross-validation setting. The classification task is to predict which of two sets of words was being read based on the respective feature representations of these words (Mitchell *et al.*, 2008; Wehbe *et al.*, 2014b,a). This task is performed between sets of 20 consecutive TRs in fMRI (accounting for the slowness of the hemodynamic response), and sets

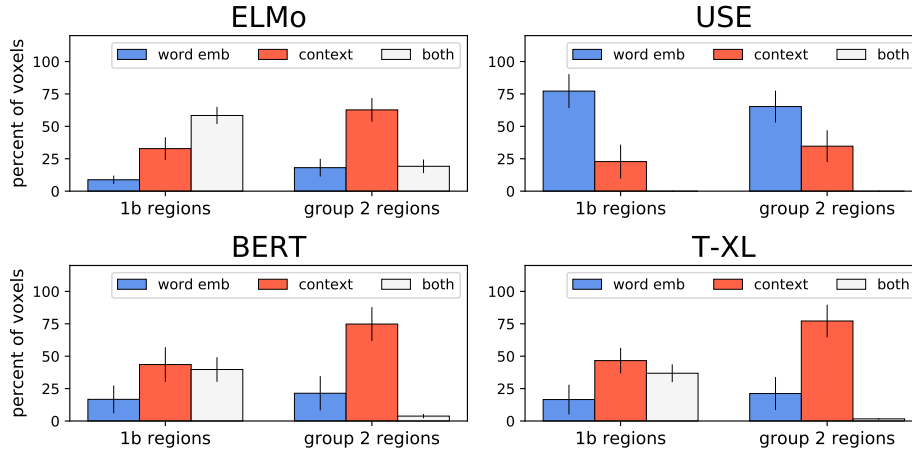

Figure 3: Amount of group 1b regions and group 2 regions predicted well by each network-derived representation: a 10-word representation corresponding to the 10 most recent words shown to the participant (Red) and a word-embedding corresponding to the last word (Blue). White indicates that both representations predict the specified amount of the regions well (about 0.7 threshold). We present the mean and standard error of the percentage of explained voxels within the specified regions over all participants.

of 20 randomly sampled words in MEG. The classification is repeated a large number of times and an average classification accuracy is obtained for each voxel in fMRI and for each sensor/timepoint in MEG. We refer to this accuracy of matching the predictions of an encoding model to the correct brain recordings as "prediction accuracy". The final fMRI results are reported on the MNI template, and we use pycortex to visualize them Gao *et al.* (2015). See the supplementary materials for more details about our methods.

**Proof of concept** Since MEG signals are faster than the rate of word presentation, they are more appropriate to study the components of word embeddings than the slow fMRI signals that cannot be attributed to individual words. We know that a word embedding learned from a text corpus is likely to contain information related to the number of letters and part of speech of a word. We show in section 4 of the supplementary materials that the number of letters of a word and its ELMo embedding predict a shared portion of brain activity early on (starting 100ms after word onset) in the back of the MEG helmet, over the visual cortex. Indeed, this region and latency are when we expect the visual information related to a word to be processed (Sudre *et al.*, 2012). Further, a word's part of speech and its ELMo embedding predict a shared portion of brain activity around 200ms after word onset in the left front of the MEG sensor. Indeed, we know from electrophysiology studies that part of speech violations incur a response around 200ms after word onset in the frontal lobe (Frank *et al.*, 2015). We conclude from these experiments that the ELMo embedding contains information about the number of letters and the part of speech of a word. Since we knew this from the onset, this experiment serves as a proof of concept for using our approach to interpret information in network representations.

## 4 Interpreting long-range contextual representations

**Integrated contextual information in ELMo, BERT, and T-XL** One question of interest in NLP is how successfully a model is able to integrate context into its representations. We investigate whether the four NLP models we consider are able to create an integrated representation of a text sequence by comparing the performance of encoding models trained with two kinds of representations: a token-level word-embedding corresponding to the most recent word token a participant was shown and a 10-word representation corresponding to the 10 most recent words. For each of the models with multiple layers (all but USE), this 10-word representation was derived from a middle layer in the network (layer 1 in ELMo, layer 7 in BERT, and layer 11 in T-XL). We present the qualitative comparisons across the four models in figure 2, where only significantly predicted voxels for each of the 8 subjects were included with the false discovery rate controlled at level 0.05 (see section 3

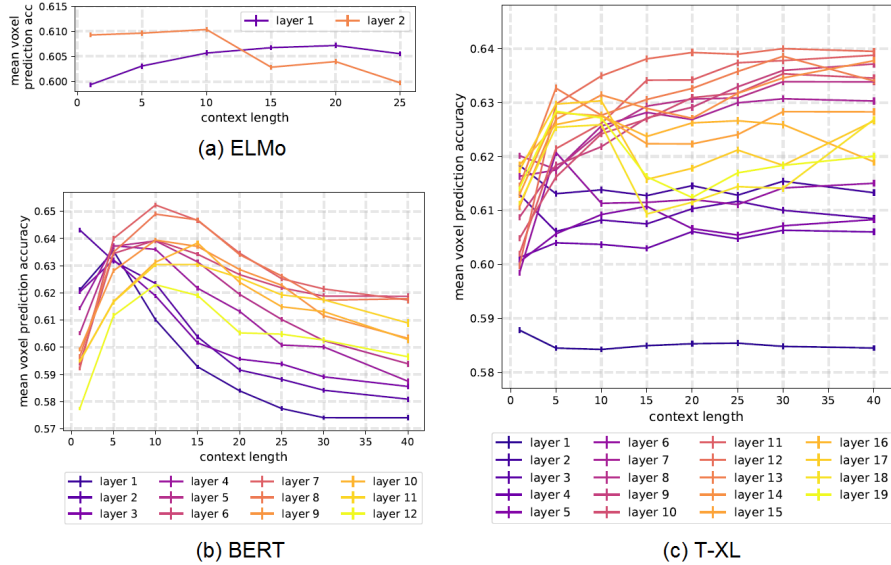

Figure 4: Performance of encoding models for all hidden layers in ELMo, BERT, and T-XL as the amount of context provided to the network is increased. Transformer-XL is the only model that continues to increase performance as the context length is increased. In all networks, the middle layers perform the best for contexts longer than 15 words. The deepest layers across all networks show a sharp increase in performance at short-range context (fewer than 10 words), followed by a decrease in performance.

of supplementary materials for more details). We provide a quantitative summary of the observed differences across models for the 1b regions and group 2 regions in Figure3. We observe similarities in the word-embedding performances across all models, which all predict the brain activity in the left and right group 1b regions and to some extent in group 1a regions. We also observe differences in the longer context representations between USE and the rest of the models:

- ELMo, BERT, and T-XL long context representations predict subsets of both group 1 regions and group 2 regions. Most parts that are predicted by the word-embedding are also predicted by the long context representations (almost no blue voxels). We conclude that the long context representations most probably include information about the long range context and the very recent word embeddings. These results may be due to the fact that all these models are at least partially trained to predict a word at a given position. They must encode long range information and also local information that can predict the appropriate word.

- USE long context representations predict the activity in a much smaller subset of group 2 regions. The low performance of the USE vectors might be due to the deep averaging which might be composing words in a crude manner. The low performance in predicting group 1 regions is most probably because USE computes representations at a sentence level and does not have the option of retaining recent information like the other models. USE long context representations therefore only have long range information.

**Relationship between layer depth and context length**   We investigate how the performances of ELMo, BERT, and T-XL change at different layers as they are provided varying size of contexts. The results are shown in figure 4. We observe that in all networks, the middle layers perform the best for contexts longer than 15 words. In addition, the deepest layers across all networks show a sharp increase in performance at short-range context (fewer than 10 words), followed by a decrease in performance. We further observe that T-XL is the only model that continues to increase performance as the context length is increased. T-XL was designed to represent long range information better than a usual transformer and our results suggest that it does. Finally, we observe that layer 1 in BERT behaves differently from the first layers in the other two networks. In figure 5, we show that when we instead examine the increase in performance of all subsequent layers from the performance of the first layer, the resulting context-layer relationships resemble the ones in T-XL. This suggests that

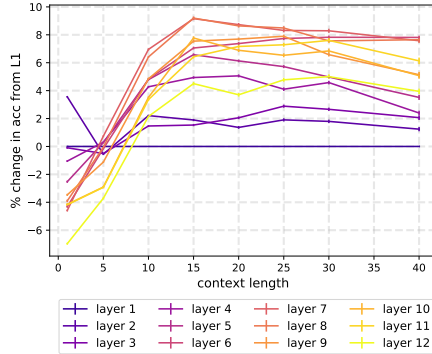 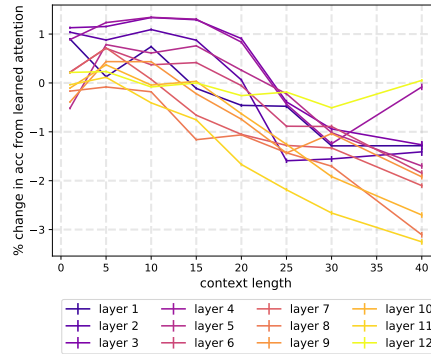

Figure 5: Change in encoding model performance of BERT layers from the performance of the first layer. When we adjust for the performance of the first layer, the performance of the remaining layers resemble that of T-XL more closely, as shown in Figure 4.

Figure 6: Change in encoding model performance of BERT layer $l$ when the attention in layer $l$ is made uniform. The performance of deep layers, other than the output layer, is harmed by the change in attention. Shallow layers benefit from the uniform attention for context lengths up to 25 words.

BERT layer 1 combines the information from the token-level embeddings in a way that limits the retention of longer context information in the layer 1 representations.

**Effect of attention on layer representation** We further investigate the effect of attention across different layers by measuring the negative impact that removing its learned attention has on its brain prediction performance. Specifically we replaced the learned attention with uniform attention over the representations from the previous layer. More concretely, to alter the attention pattern at a single layer in BERT, for each attention head $h_i = Attn_i(QW_i^Q, KW_i^K, VW_i^V)$, we replace the pretrained parameter matrices $W_i^Q$, $W_i^K$, and $W_i^V$ for this layer, such that the attention $Attn(Q, K, V)$, defined as $softmax(QK/\sqrt{d_k})^T V$ (Vaswani *et al.*, 2017), yields equal probability over the values in value matrix $V$ (here $d_k$ denotes the dimensionality of the keys and queries). To this end, for a single layer, we replace $W_i^Q$ and $W_i^K$ with zero-filled matrices and $W_i^V$ with the identity matrix. We only alter a single layer at a time, while keeping all other parameters of the pretrained BERT fixed. In figure 6, we present the change in performance of each layer with uniform attention when compared to pretrained attention. The performance of deep layers, other than the output layer, is harmed by the change in attention. However, surprisingly and against our expectations, shallow layers benefit from the uniform attention for context lengths up to 25 words.

## 5 Applying insight from brain interpretations to NLP tasks

After observing that the layers in the first half of the base BERT model benefit from uniform attention for predicting brain activity, we test how the same alterations affect BERT's ability to predict language by testing its performance on natural language processing tasks. We evaluate on tasks that do not require fine-tuning beyond pretraining to ensure that there is an opportunity to transfer the insight from the brain interpretations of the pretrained BERT model. To this end, we evaluate on a range of syntactic tasks proposed by Marvin and Linzen (2018), that have been previously used to quantify BERT's syntactic capabilities (Goldberg, 2019). These syntactic tasks measure subject-verb agreement in various types of sentences. They can be thought of as probe-tasks because they assess the ability of the network to perform syntax-related predictions without further fine-tuning.

We adopt the evaluation protocol of Goldberg (2019), in which BERT is first fed a complete sentence where the single focus verb is masked (e.g. `[CLS] the game that the guard hates [MASK] bad .`), then the prediction for the masked position is obtained using the pretrained language-modeling head, and lastly the accuracy is obtained by comparing the scores for the original correct verb (e.g. `is`) to the score for the incorrect verb (i.e. the verb that is wrongly numbered) (e.g. `are`). We make the attention in layers 1 through 6 in base BERT uniform, a single

| condition | uni L1 | uni L2 | uni L6 | uni L11 | base | count |
|---|---|---|---|---|---|---|
| simple | **1.00** | **1.00** | **1.00** | 0.98 | **1.00** | 120 |
| in a sentential complement | **0.83** | **0.83** | **0.83** | **0.83** | **0.83** | 1440 |
| short VP coordination | 0.88 | 0.90 | **0.91** | 0.88 | 0.89 | 720 |
| long VP coordination | 0.96 | 0.97 | **1.00**** | 0.96 | 0.98 | 400 |
| across a prepositional phrase | 0.86 | **0.93**** | 0.88 | 0.82 | 0.85 | 19440 |
| across a subject relative clause | 0.83 | 0.83 | **0.85**** | 0.83 | 0.84 | 9600 |
| across an object relative clause | 0.87 | 0.91 | **0.92**** | 0.86 | 0.89 | 19680 |
| across an object relative clause (no that) | **0.87** | 0.80 | **0.87** | 0.84 | 0.86 | 19680 |
| in an object relative clause | **0.97**** | 0.95 | 0.91 | 0.93 | 0.95 | 15960 |
| in an object relative clause (no that) | **0.83**** | 0.72 | 0.74 | 0.72 | 0.79 | 15960 |
| reflexive anaphora: simple | 0.91 | 0.94 | **0.99**** | 0.95 | 0.94 | 280 |
| reflexive anaphora: in a sent. complem. | 0.88 | 0.85 | 0.86 | 0.85 | **0.89** | 3360 |
| reflexive anaphora: across rel. clause | 0.79 | **0.84**** | 0.79 | 0.76 | 0.80 | 22400 |

Table 1: Performance of models with altered attention on subject-verb agreement across various sentence types (tasks by Marvin and Linzen (2018)). Best performance per task is made bold, and marked with ** when difference from 'base' performance is statistically significant. The altered models for the shallow layers significantly outperform the pretrained model ('base') in 8 of the 13 tasks and achieve parity in 4 of the remaining 5 tasks.

layer at a time while keeping the remaining parameters fixed as described in Section 4, and evaluate on the 13 tasks. We present the results of altering layers 1,2, and 6 in Table 1. We observe that the altered models significantly outperform the pretrained model ('base') in 8 of the 13 tasks and achieve parity in 4 of the remaining 5 tasks (paired t-test, significance level 0.01, FDR controlled for multiple comparisons (Benjamini and Hochberg, 1995)). Performance of altering layers 3-5 is similar and is presented in Supplementary Table 2. We contrast the performance of these layers with that of a model with uniform attention at layer 11, which is the model that suffers the most from this change for predicting the brain activity as shown in Figure 6. We observe that this model also performs poorly on the NLP tasks as it performs on par or worse than the base model in 12 of the 13 tasks.

## 6 Discussion

We introduced an approach to use brain activity recordings of subjects reading naturalistic text to interpret different representations derived from neural networks. We used MEG to show that the (non-contextualized) word embedding of ELMo contains information about word length and part of speech as a proof of concept. We used fMRI to show that different network representation (for ELMo, USE, BERT, T-XL) encode information relevant to language processing at different context lengths. USE long-range context representations perform differently from the other model and do not also include short-range information. The transformer models (BERT and T-XL) both capture the most brain-relevant context information in their middle layers. T-XL, by combining both recurrent properties and transformer properties, has representations that don't degrade in performance when very long context is used, unlike purely recurrent models (e.g. ELMo) or transformers (e.g. BERT).

We found that uniform attention on the previous layer actually improved the brain prediction performance of the shallow layers (layers 1-6) over using learned attention. After this observation, we tested how the same alterations affect BERT's ability to predict language by probing the altered BERT's representations using syntactic NLP tasks. We observed that the altered BERT performs better on the majority of the tasks. This finding suggests that altering an NLP model to better align with brain recordings of people processing language may lead to better language understanding by the NLP model. This result parallels concurrent work by Kubilius *et al.* (2019) in the domain of vision, which shows that a neural network that is better-aligned with brain activity and incorporates insights about modularity and connectivity in the brain outperforms other models with similar capacity on Imagenet.

**Future work** We hope that as naturalistic brain experiments become more popular and data more widely shared, aligning brain activity with neural network will become a research area. Our next steps are to expand the analysis using MEG to uncover new aspects of word-embeddings and to derive more informative fMRI brain priors that contain specific conceptual information that is linked to brain areas, and use them to study the high level semantic information in network representations.

**Acknowledgments**

We thank Tom Mitchell for valuable discussions. We thank the National Science Foundation for supporting this work through the Graduate Research Fellowship under Grant No. DGE1745016, and Google for supporting this work through the Google Faculty Award.

## Footnotes

[1]https://github.com/huggingface/pytorch-pretrained-BERT/

[2]https://tfhub.dev/google/universal-sentence-encoder/2

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
