[Supplementary Material]

# Supplementary Materials: Interpreting and improving natural-language processing (in machines) with natural language-processing (in the brain)

**Mariya Toneva**
Neuroscience Institute
Department of Machine Learning
Carnegie Mellon University
mariya@cmu.edu

**Leila Wehbe**
Neuroscience Institute
Department of Machine Learning
Carnegie Mellon University
lwehbe@cmu.edu

## 1  Brain areas included in prior

| | |
|---|---|
| 1a | Inferior Frontal Gyrus |
| 1b | Middle/Superior Temporal |
| 2a | Lateral Middle/Superior Frontal |
| 2b | Supramarginal Gyrus / Posterior Superior Temporal / Angular Gyrus |
| 2c | Precuneus |
| 2d | Medial Superior Frontal |
| 2e | Medial Orbito-Frontal |

Table 1: Name of regions of interest in fig. 1 of main manuscript. Regions were approximated from the results of (Lerner *et al.*, 2011).

## 2  Data Preprocessing

We use fMRI data of 8 subjects reading chapter 9 of *Harry Potter and the Sorcerer's Stone* (Rowling, 2012), collected and made available online by Wehbe *et al.* (2014b)[1]. Words were presented one at a time at a rate of 0.5s each. fMRI data was acquired at a rate of 2s per image, i.e. the repetition time (TR) is 2s. The images were comprised of $3 \times 3 \times 3mm$ voxels. The data for each subject was slice-time and motion corrected using SPM8 (Kay *et al.*, 2008), then detrended and smoothed with a 3mm full-width-half-max kernel. The brain surface of each subject was reconstructed using Freesurfer (Fischl, 2012), and a grey matter mask was obtained. The Pycortex software (Gao *et al.*, 2015) was used to handle and plot the data. For each subject, 25000-31000 cortical voxels were kept.

The same paradigm was recorded for 3 subjects using MEG by the authors of Wehbe *et al.* (2014a) and shared upon our request. This data was recorded at 306 sensors organized in 102 locations around the head. MEG records the change in magnetic field due to neuronal activity and the data we used was sampled at 1kHz, then preprocessed using the Signal Space Separation method (SSS) (Taulu *et al.*, 2004) and its temporal extension (tSSS) (Taulu and Simola, 2006). The signal in every sensor was downsampled into 25ms non-overlapping time bins. For each of the 5176 word in the chapter, we therefore obtained a recording for 306 sensors at 20 time points after word onset (since each word was presented for 500ms).

# 3 Encoding Models

## 3.1 fMRI

Ridge regularization is used to estimate the parameters of a linear model that predicts the brain activity $y^i$ in every fMRI voxel $i$ as a linear combination of a particular layer representation $x^\ell$. For each output dimension (voxel), the Ridge regularization parameter is chosen independently by nested cross-validation. We use Ridge regression because of its computational efficiency and because of the results of Wehbe *et al.* (2015) showing that for fMRI data, as long as proper regularization is used and the regularization parameter is chosen by cross-validation for each voxel independently, different regularization techniques lead to similar results. Indeed, Ridge regression is indeed a common regularization technique used for building predictive fMRI (Mitchell *et al.*, 2008; Nishimoto *et al.*, 2011; Wehbe *et al.*, 2014b; Huth *et al.*, 2016).

For every voxel $i$, a model is fit to predict the signals $y^i = [y_1^i, y_2^i, \ldots, y_n^i]$, where $n$ is the number of time points, as a function of the representation derived from layer $\ell$ of a network. The words presented to the participants are first grouped by the TR interval in which they were presented. Then, the features of layer $\ell$ of the words in every group are averaged to form a sequence of features $x^\ell = [x_1^\ell, x_2^\ell, \ldots, x_n^\ell]$ which are aligned with the brain signals. The models are trained to predict the signal at time $t$, $y_t$, using the concatenated vector $z_t^\ell$ formed of $[x_{t-1}^\ell, x_{t-2}^\ell, x_{t-3}^\ell, x_{t-4}^\ell]$. The features of the words presented in the previous volumes are included in order to account for the lag in the hemodynamic response that fMRI records. Indeed, the response measured by fMRI is an indirect consequence of brain activity that peaks about 6 seconds after stimulus onset, and the solution of expressing brain activity as a function of the features of the preceding time points is a common solution for building predictive models (Nishimoto *et al.*, 2011; Wehbe *et al.*, 2014b; Huth *et al.*, 2016).

For each given subject and each layer $\ell$, we perform a cross-validation procedure to estimate how predictive that layer is of brain activity in each voxel $i$. For each fold:

- The fMRI data $Y$ and feature matrix $Z^\ell = z_1^\ell, z_2^\ell, \ldots z_n^\ell$ are split into corresponding train and validation matrices and these matrices are individually normalized (to get a mean of 0 and standard deviation of 1 for each voxel across time), ending with train matrices $Y^R$ and $Z^{R,\ell}$ and validation matrices $Y^V$ and $Z^{V,\ell}$.

- Using the train fold, a model $w^{i,\ell}$ is estimated as:

$$\arg \min_{w^{i,\ell}} ||y^{R,i} - Z^{R,\ell} w^{i,\ell}|_2^2 + \lambda^i ||w^{i,\ell}||_2^2$$

  A ten-fold nested cross-validation procedure is first used to identify the best $\lambda^i$ for every voxel $i$ that minimizes nested cross-validation error. $w^{i,\ell}$ is then estimated using $\lambda^i$ on the entire training fold.

- The predictions for each voxel on the validation fold are obtained as $p^\ell = Z^{V,\ell} w^{i,\ell}$.

- A classification task is then performed to assess the prediction performance of the learned model. This classification task is based on searchlight classification (Kriegeskorte *et al.*, 2006), in which a sliding window groups each voxel with its immediate neighbors in the 3D grid of voxels. We perform a more accurate searchlight analysis we refer to as cortical-searchlight. We are interested only in the grey matter voxels (which contain neurons) and these comprise the most external part of the brain: the cortical sheet. The cortical sheet of each subject is highly folded, and voxels that lie in a neighborhood on the sheet are not necessarily neighbors in the 3D grid of voxels. Using the reconstructed cortical sheet of each subject, we estimate for each cortical voxel a surrounding neighborhood by including the voxels adjacent to it on the cortical sheet, and the voxels adjacent to those voxels. See figure 1. We use for each voxel $i$ this neighborhood of voxels $N^i$ with $|N^i| = k^i$ in a classification task.

- For each voxel $i$, we use the signals predicted for layer $\ell$ to classify a contiguous chunk of real data of length 20TRs. Since fMRI data is noisy, performance using a single TR will be close to chance accuracy and will therefore have low power and will not be informative for our purpose. Indeed, for this reason most experiments using predictive fMRI models test

Figure 1: Example neighborhood estimated using the cortical sheet and not the 3D grid of voxels.

them on a part of the experiment that is repeated multiple times (Kay *et al.*, 2008; Nishimoto *et al.*, 2011; Huth *et al.*, 2016). These repetitions are then averaged into one test set which is predicted, and this less noisy average leads to better prediction accuracy. The experiment we are using however doesn't have any repetitions and not specific test set, and therefore by raise the number of TRs and classify 20TRs at a time, we are able to improve the classification accuracy. Wehbe *et al.* (2014b) have shown that classification accuracy reaches a plateau after around 15 TRs and we pick 20TRs for good measure. The classification task takes an unlabeled chunk of real data of size $20 \times k$ and two possible predicted data chunks of the same size, one being the predicted data corresponding to the same time, and another randomly chosen chunk. Euclidean distance is computed between the real chunk and the two predicted chunks, and the closest chunk is chosen. This is repeated a large number of times and average accuracy is computed at each voxel.

The above steps are repeated for each of the four cross-validation folds and average accuracy is obtained for each voxel $i$ for layer $\ell$, for each subject.

We use a new empirical based method to compute statistical significance that relies on the distribution of average accuracies over a subject's brain to estimate the False Discovery Proportion (FDP). The voxel accuracies belong to two distributions: either the voxel has chance accuracy or the voxel is truly predicted by the corresponding layer $\ell$. Average chance accuracy for our binary balanced task is 0.5, however the accuracies due to chance performance might have a varying distribution around 0.5. The accuracies above 0.5 are a mixture of predicted voxels and voxels with chance performance. We assume that chance performance is symmetrically distributed around 0.5, and we use the set of accuracies that are less than 0.5–which we consider to be in the chance distribution–to estimate the distribution of chance accuracies above 0.5. We want to find a set of voxels where to reject the null hypothesis such that the FDP is $\leq 0.05$. For that purpose we find the smallest margin $\delta$, $0 < \delta < 0.5$ such that:

$$\widehat{\mathrm{FDP}} = \frac{1 + \#\{\text{voxel } s.t. \text{ accuracy} \leq 0.5 - \delta\}}{1 \vee \#\{\text{voxel } s.t. \text{ accuracy} \geq 0.5 + \delta\}} \leq q$$

where $q = 0.05$, by starting at $\delta = 0.001$ and increasing it in increments of 0.001, stopping when $\widehat{\mathrm{FDP}} \leq 0.05$ or the limit is reached. This approach is adapted from the Barber-Candès approach which has been proposed and analyzed by Barber *et al.* (2015); Arias-Castro *et al.* (2017); Rabinovich *et al.* (2017), and shown to control the False Discovery Rate (FDR) at level $q$ when $\delta_{\text{final}}$ is chosen as a threshold. We reject the null hypothesis for all voxels where the accuracy is $\geq 0.5 + \delta_{\text{final}}$.

To combine results across different subjects, we use pycortex (Gao *et al.*, 2015) to transform each subject to the Montreal Neurological Institute (MNI) space, the most commonly used template space in fMRI. We can then average the results of different participants.

## 3.2 MEG

MEG data is sampled faster than the rate of word presentation, so for each word, we have 20 times points recorded at 306 sensors. Ridge regularization is similarly used to estimate the parameters of a

linear model that predicts the brain activity $y^{i,\tau}$ in every MEG sensor $i$ at time $\tau$ after word onset. For each output dimension (sensor/time tuple $i, \tau$), the Ridge regularization parameter is chosen independently by nested cross-validation.

For every sensor/time tuple $i, \tau$, a model is fit to predict the signals $y^{i,\tau} = [y_1^{i,\tau}, y_2^{i,\tau}, \ldots, y_n^{i,\tau}]$, where $n$ is the number of words in the story, as a function of the representation derived from layer $\ell$ of a network. We use as input the word vector $x^\ell$ without the delays we used in fMRI because the MEG recordings capture instantaneous consequences of brain activity (change in the magnetic field). The models are trained to predict the signal at word $t$, $y_t^{i,\tau}$, using the vector $x_t^\ell$.

For each each given subject and each layer $\ell$, we perform a cross-validation procedure to estimate how predictive that layer is of brain activity in each voxel $i$. For each fold:

- The MEG data $Y$ and feature matrix $X^\ell = x_1^\ell, x_2^\ell, \ldots x_n^\ell$ are split into corresponding train and validation matrices and these matrices are individually normalized (to get a mean of 0 and standard deviation of 1 for each voxel across time), ending with train matrices $Y^R$ and $X^{R,\ell}$ and validation matrices $Y^V$ and $Z^{V,\ell}$.

- Using the train fold, a model $w^{(i,\tau)\ell}$ is estimated as:

$$\arg \min_{w^{(i,\tau)\ell}} ||y^{(i,\tau),R} - X^{R,\ell} w^{(i,\tau)\ell}|_2^2 + \lambda^{(i,\tau)} ||w^{(i,\tau)\ell}||_2^2$$

  A ten-fold nested cross-validation procedure is first used to identify the best $\lambda^{(i,\tau)}$ for every sensor, time-point tuple $(i, \tau)$ that minimizes nested cross-validation error. $w^{(i,\tau)\ell}$ is then estimated using $\lambda^{(i,\tau)}$ on the entire training fold.

- The predictions for each sensor, time-point tuple $(i, \tau)$ on the validation fold are obtained as $p^\ell = X^{V,\ell} w^{(i,\tau)\ell}$.

- A classification task is then performed to assess the prediction performance of the learned model. This classification task also pools spatially: we use the 3 sensors at each location, pooling across all the subjects, ending up with 102 classifications at 20 time-points. By pooling the data in each sensor location across subjects, we increase the signal-to-noise ratio.

- For each sensor location $s$ and time-point $\tau$, we use the signals predicted from layer $\ell$ for the three sensors at time-point $\tau$ after word onset to classify a set of 20 words. Since MEG data is noisy, performance using a single word will be close to chance accuracy and will therefore have low power and will not be informative for our purpose. Indeed, for this reason most experiments using predictive MEG models test them on a part of the experiment that is repeated multiple times (Sudre *et al.*, 2012). These repetitions are then averaged into one test set which is predicted, and this less noisy average leads to better prediction accuracy. The experiment we are using however doesn't have any repetitions and not specific test set, and therefore by raising the number of words and classify 20 words at a time, we are able to improve the classification accuracy. We use the value of 20 words from Wehbe *et al.* (2014a).

The above steps are repeated for each of the four cross-validation folds and average accuracy is obtained for each sensor location, time-point tuple $(s, \tau)$ for layer $\ell$, for each subject.

In our proof of concept experiment, we run an analysis in which we try to find, using the classification task outlined here, classification accuracy that is common both to a word embedding $\ell$ and to other features of a word such as a one-hot vector encoding its part of speech. This analysis is a proxy for finding the shared explained variance between the vectors, which we can call A and B. We concatenate A and B into a vector (representing $A \cup B$). We run the classification analysis using $A$, $B$ and $(A \cup B)$. We then estimate the shared accuracy as: $A + B - A \cup B$.

## 4 MEG results as proof of concept

We use MEG to provide a proof of concept of our approach. We know that single word non-contextualized embeddings likely have information about the part-of-speech and the length of a

Figure 2: Performance of ELMo current word embedding at predicting MEG activity at each sensor location and time point, compared with the performance shared with word length and Part-Of-Speech (POS) tags. Around 200-250ms, the word embedding predicts a part of the activity at the top of the helmet, and this is shared mostly with the POS tags and not with word length (see bottom-right comparison). Indeed, we know from electrophysiology studies studies that POS violations incur a response around 200ms after word onset in the front of the brain Frank *et al.* (2015), which aligns with our analysis. We hypothesize from these results that the word-embedding contains both word length and POS information.

word. We will show here how our approach can recover this information from brain activity as a proof-of-concept. We use MEG to study word embeddings because unlike fMRI we can access the brain activity to reading a single word. We know from the Neuroscience literature that MEG activity can be related to the length of the current word Sudre *et al.* (2012) and its part of speech Frank *et al.* (2015) at different times. We investigate whether word length and part-of-speech (POS) information is also present in the non-contextualized embedding by computing the shared performance ($A \cap B$) between the pairs of features ($A$ and $B$) as $A + B - A \cup B$ as explained in the previous section.

We present the results in Figure 2. The current word embedding is able to predict activity as the current word is being perceived starting at the back of the sensor helmet (generally on top of the visual cortex) around 100ms. This is when we expect the visual signal to start reaching the visual cortex. Indeed, we see that the word-embedding and the word length have overlap in the activity they predict in the visual cortex at that time. Gradually, the areas predicted by the word embedding move forward in the brain towards areas known to be involved in more high level aspects of reading. Around 200-250ms, we see the word embedding predicts a part of the activity at the top of the helmet, and this is shared mostly with the POS tags and not with word length (see bottom-right comparison). Indeed, we know from electrophysiology studies studies that POS violations incur a response around 200ms after word onset in the front of the brain Frank *et al.* (2015), which aligns with our analysis. From these results we can hypothesize that the word-embedding contains both word length and POS information, as was expected.

# 5 Complete Attention Results

| condition | uni L1 | uni L2 | uni L3 | uni L4 | uni L5 | uni L6 | base | count |
|---|---|---|---|---|---|---|---|---|
| simple | 1.00 | 1.00 | 0.96 | 1.00 | 0.99 | 1.00 | 1.00 | 120 |
| in a sentential complement | 0.83 | 0.83 | 0.83 | 0.83 | 0.84 | 0.83 | 0.83 | 1440 |
| short VP coordination | 0.88 | 0.90 | 0.91 | 0.88 | 0.88 | 0.91 | 0.89 | 720 |
| long VP coordination | 0.96 | 0.97 | 0.95 | 0.95 | 0.96 | 1.00 | 0.98 | 400 |
| across a prepositional phrase | 0.86 | 0.93 | 0.88 | 0.86 | 0.80 | 0.88 | 0.85 | 19440 |
| across a subject relative clause | 0.83 | 0.83 | 0.84 | 0.84 | 0.83 | 0.85 | 0.84 | 9600 |
| across an object relative clause | 0.87 | 0.91 | 0.90 | 0.86 | 0.83 | 0.92 | 0.89 | 19680 |
| across an object relative clause (no that) | 0.87 | 0.80 | 0.75 | 0.72 | 0.75 | 0.87 | 0.86 | 19680 |
| in an object relative clause | 0.97 | 0.95 | 0.96 | 0.92 | 0.91 | 0.91 | 0.95 | 15960 |
| in an object relative clause (no that) | 0.83 | 0.72 | 0.70 | 0.69 | 0.74 | 0.74 | 0.79 | 15960 |
| reflexive anaphora: simple | 0.91 | 0.94 | 0.99 | 0.98 | 1.00 | 0.99 | 0.94 | 280 |
| reflexive anaphora: in a sent. complem. | 0.88 | 0.85 | 0.88 | 0.87 | 0.86 | 0.86 | 0.89 | 3360 |
| reflexive anaphora: across a rel. clause | 0.79 | 0.84 | 0.82 | 0.68 | 0.66 | 0.79 | 0.80 | 22400 |

Table 2: Performance of models with uniformly-altered attention in layers 1-6 in BERT on a range of syntactic tasks by Marvin and Linzen (2018). 'Base' refers to pretrained BERT.

## Footnotes

[1]http://www.cs.cmu.edu/afs/cs/project/theo-73/www/plosone/