[Reviews · NeurIPS 2019]

Reviewer 1



The paper uses brain activity data (fMRI and MEG) obtained from subjects while reading natural text and computes representations of NN models (ELMO, BERT, etc.) on the same text data. The goal is to see which layers predict brain activity in different areas of the brain, as well as the role of context size for each of the method. Conclusions: - T-XL increase prediction accuracy with increase of context size - BERT and T-XL capture context in a way that is relevant to predicting brain activity in their middle layers - Removing the attention at various layers for BERT has similar effects on brain prediction and NLP tasks: lower layer uniform attention for BERT is better in both cases, while at layer 11 it decreases performance in both cases. While this type of analysis is not completely novel, the observations made are new and very interesting. For the most part the paper is clearly written (See Improvements section for clarification questions). The paper would be stronger if the empirical implications of the observations (the attention removal) would be tested more.

Reviewer 2



This paper presents interesting work on using brain imaging data to measure the quality of linguistic representations. Although the authors claim that they are the first in doing this, I believe there is a long tradition with this scope. See, for instance, https://aclweb.org/anthology/papers/D/D13/D13-1202/. Nonetheless, this is important work. I can't judge for the neuroscience methods, but otherwise the paper seems pretty solid. There are a few points that I'm not completely sure about: * The authors explore a fixed length window, whereas many of these models are trained at the sentence level. Aren't they introducing some arbitrary artifacts there? * The data reported in Figure 2 may be also interesting to report in a quantitative fashion (like, the distribution of red/blue areas in regions 1 and 2). * What are called NLP tasks, they are more like syntactic processing diagnostic tasks. Probably worth distinguishing them from downstream NLP tasks (like, sentiment analysis)

Reviewer 3



This work describes experiments done using pretrained word embedding model representations, and fMRI and MEG representations when reading the same text. Bringing these together is original and an interesting avenue of research, yet I have doubts about the significance, clarity and quality of this work. There are several points where references would have been needed to refer to prior work, or to back up some claim (e.g. line 41, line 63). The paper furthermore has a significant part of its material deferred into the appendix, including parts that are crucial to understand the experiment. From the main paper alone it is for example unclear which metrics are used when evaluating the fitted linear models, and even information as basic as whether the task is a regression or a classification task. In the main prediction task, no other baselines are tested (e.g. prediction from previous brain activity alone, without any text encoder representations). Linking observations made with the brain activity prediction model (uniform attention) to better NLP task performance is a relatively weak argument that would in my view need additional justification or empirical support. I am not convinced that the presented results (Table 1) are clearly different from the base model (small sample sizes, multiple testing) nor that the chosen syntactic tasks are very meaningful for NLP tasks in general. I also believe that predictiveness for other tasks than brain activity could equally have suggested the uniform-attention layer modifications, and one would have to be more precise in what exactly predicting fMRI or MEG adds, as compared to predictiveness of other NLP signals (e.g. syntactic). Changing the BERT model architecture after pretraining is somewhat of a hack -- the model would have to be retrained again with the new architecture to test whether the claimed insights on architecture have a practical benefit. I did not obtain insight or interpretation of the pretrained neural models, as suggested in the abstract ("We propose here a novel approach for interpreting neural networks [...]"), to the point that I think this is misleading. There are very specific hypotheses made: "When we align that specific network representation with fMRI and MEG data, the result will be a decomposition of the representation into parts that correspond to different processes and should therefore be more interpretable." (line 67) which are neither experimentally tested nor referred back to later on. Overall I think this is a very interesting direction of research, the paper is well-related to prior work, and takes care about some important experimental aspects (cross-validation, multiple participants, etc.). But overall the line of argumentation, presentation, and experiments have not convinced me.

[Author Response · NeurIPS 2019]

We thank the reviewers for the thoughtful comments and attempt to address their questions, space permitting.

**All reviewers:** R1 and R2 correctly point out that relating intermediate representations in neural networks to brain activity has been previously explored (as we also state in L53-54). However, previous works make key untested assumptions about information contained in the neural network representations and use these representations to examine where/when this information is present the brain. Our work is the first, to our knowledge, to propose using brain activity for examining how this assumed information alters as the network representations change. While the brain has successfully informed computer vision (e.g. the hierarchy of CNNs is inspired from the visual system), the NLP field remains less convinced of the potential of the brain. We propose a framework to start changing this status-quo.

**R1 and R3:** The learned function $f$ is evaluated in a classification task on held-out data using 4-fold cross-validation (L182-185). The classification task is to predict which of 2 sets of words was being read by the participant, which is a previously established metric. Our evaluation metric is the accuracy of this classification for each voxel (theoretical chance is 0.5), summarized by the mean and standard error across voxels (Fig 3). We will clarify this in the main text.

The motivation behind the attention removal experiments was to investigate how important the learned attention is in different layers. Following one of our more surprising findings that removing attention in shallow layers can actually improve brain activity prediction, we wanted to test the same network modification in an independent NLP setting.

**R1:** Due to typically limited data, we have observed that training non-linear predictive models for fMRI that outperform linear ones is difficult. Predicting fMRI directly from language networks is nonetheless interesting future work.

**R2:** To test the effect of the fixed-length evaluation window, we additionally computed ELMo features for word $w_t$ by passing words $w_1, ..w_t$ from the sentence in which $w_t$ appears through pretrained ELMo. We found that predictions of fMRI activity using sentence features correlate strongly with predictions from ELMo representations obtained from a fixed window longer than 5 words ($0.8 \pm 0.01$ mean Pearson correlation across subjects).

Following R2's suggestion, we computed the percentages of voxels within the ROIs that are well explained by word embeddings (blue), long-context represen- tations (red), or both (white). These quantitative results (mean percentages over subjects for ELMo, shown to the right) further validate our conclusions that both word embeddings and long-context can explain 1b regions well, while group 2 regions are best predicted by long-context. We will include plots for all models.

**R3:** We will add more references to ground our statements about the brain. We address our hypothesis that a neural representation can be decomposed across time points and locations in the brain through aligning with fMRI and MEG in two experiments. First is a proof of concept (L188), showing that the ELMo word embedding aligns with times and locations in MEG corresponding to known processing of word length and part-of-speech. These are expected properties of word embeddings that can be tested with more traditional methods, and we wanted to verify that our method is also able to expose these. The second experiment was to contrast a word embedding with sequence embeddings from 4 different NLP models in their abilities to align with different locations that are known to processes single-word information and long-range context information to different extents (results in Fig 2). We agree that there is more to be done to fully characterize the many types of information contained in a neural embedding in future work.

We agree that there are other tasks that could have led to removing attention. However, we argue that predicting brain activity is more informative as it provides additional insights, such as a decomposition of the neural representation across the brain. Further, the attention experiment supports our premise that similarity of language representations to brain activity is useful, and that it reveals what representations are more relevant for language tasks, a fact we can capitalize on in future research. While retraining with the modified architecture is a good next step, the current setup tests whether the modified representations themselves, before retraining, have more language relevant information.

The syntactic tasks measure subject-verb agreement, so the "incorrect verb" is the wrongly-numbered correct verb (e.g. incorrect verb is "are" if the correct verb is "is", as in the example in L268). We will clarify this in Section 5. Following R3's suggestion, we tested the significance of the differences in accuracy on each task in Table 1 between the base model and the uniform-attention models. The uniform-attention models in the early layers presented in Table 1 (L1, L2, L6) significantly outperform the base model (paired t-test, significance level 0.01, FDR controlled for multiple comparisons) in 8 of the 13 tasks. The two numerical improvements in Table 1 that do not survive the statistical test are for the tasks "short VP coordination" and "across an object relative clause (no that)". We will indicate this in Table 1.

The delay in fMRI is due to the hemodynamic response. We account for it by building predictive models with features from words occurring in previous time points, which is a common way to correct for the delay [Nishimoto 2011 Cur. Biol.,Huth 2016 Nature], and in this way we avoid any confounding from the delay (section 3.1 of supplementary which we can add to main text). MEG does not suffer from such latency, as information due to the current word is detected in the recordings within 100 milliseconds after word presentation [Pulvermüller 2011 EJNeur].

[Meta-Review · NeurIPS 2019]

This is a well-written and thorough paper, where the idea is to compare recent successful neural-network models for language processing (both pure classification and transfer/representation-learning models) with human brain data. This is an important goal, because humans have neural networks that can make sense of language, and there may be a lot that the machine learning community can learn from understanding these exemplary language processors better. Indeed, a teleological explanation of how human or artificial neural-networks process language would be an enormous breakthrough in language science. The reviewers are positive about this paper and therefore I support them in recommending acceptance. To add to the positive aspects that they point out, I have a few concerns about the framing, which I list below in case they can improve the camera ready submission: 1) The abstract states that: "it is still unclear what the representations learned by these networks correspond to". It seems to me that this paper does not really answer the question it poses in the first line of the abstract. I would recommend revising this rhetoric, because I don't think the paper shows what the representations learned by big neural networks correspond to (at least in terms of things in language or things in the world). Even if we could determine what patterns of activation in deep neural networks correspond to it 's not clear that that would provide a causal explanation of how the network computes predictions or behaviour given stimuli. This is question that much neuroscience has engaged with, and I feel that it could be brought out by the authors in this work. Certainly identifying correlations, which is the approach of this paper, is very different from helping to explain something. According to the scientific method, explaining something should really involve a 'theory' of how the model is doing something, followed by a hypothesis test. 2) The paper claims to 'improve and interpret' NLP models (or BERT), but I'm still unsure whether it does the claim to improve BERT is fair. The improvement that is exhibited is on a very esoteric task, that is more like a probe task than a practical application. To show that BERT has been been improved should probably involve evaluations on the standard tasks on which BERT was originally evaluated.